# Zebrafish as a Successful Animal Model for Screening Toxicity of Medicinal Plants

**DOI:** 10.3390/plants9101345

**Published:** 2020-10-12

**Authors:** Amir Modarresi Chahardehi, Hasni Arsad, Vuanghao Lim

**Affiliations:** Integrative Medicine Cluster, Advanced Medical and Dental Institute, Universiti Sains Malaysia, Bertam, Kepala Batas 13200, Malaysia; hasniarsad@usm.my

**Keywords:** zebrafish, toxicity, embryotoxicity, medicinal plant, animal model

## Abstract

The zebrafish (*Danio rerio*) is used as an embryonic and larval model to perform in vitro experiments and developmental toxicity studies. Zebrafish may be used to determine the toxicity of samples in early screening assays, often in a high-throughput manner. The zebrafish embryotoxicity model is at the leading edge of toxicology research due to the short time required for analyses, transparency of embryos, short life cycle, high fertility, and genetic data similarity. Zebrafish toxicity studies range from assessing the toxicity of bioactive compounds or crude extracts from plants to determining the optimal process. Most of the studied extracts were polar, such as ethanol, methanol, and aqueous solutions, which were used to detect the toxicity and bioactivity. This review examines the latest research using zebrafish as a study model and highlights its power as a tool for detecting toxicity of medicinal plants and its effectiveness at enhancing the understanding of new drug generation. The goal of this review was to develop a link to ethnopharmacological zebrafish studies that can be used by other researchers to conduct future research.

## 1. Introduction

Herbal plants have pharmacological and therapeutic characteristics due to the natural chemical compounds they contain [1]. Hence, they are widely utilized every day for culinary purposes and nutritional supplements to promote health [2]. This product can be toxic, so its toxicity must be measured to ensure adequate safety for human health [3]. Although many people view most medicinal plants as safe, poisoning can occur in some cases. Consumers can also be exposed to potential health risks caused by specific components or contaminants of botanical products; thus, their risk needs to be evaluated [4,5]. Plant materials and their extracts contain various toxic substances synthesized by plants as a defense against disease, insects, and other organisms [6]. Botanical toxicity studies are complicated due to expense, time, use of animals, and the complex mix of components [4,7]. The most commonly used extraction solvents, from polar to non-polar, are water, ethanol, methanol, acetone, ethyl acetate, chloroform, dichloromethane and hexane, and numerous researchers have studied the effectiveness of these solvents [8]. Various extracts may display distinctive cytotoxicity traits and a wide variety of pharmacological consequences at different concentrations [9]. Additionally, numerous factors affect compounds, such as the type and volume of extraction solvents used and varying storage environments [8]. Bioactive compounds in medicinal plants can have toxic impacts on cardiac glycosides, phorbol esters, alkaloids, cyanogenic glycosides, and lectins [10]. The statistical evaluation carried out by the National Pharmaceutical Control Bureau and Health ministry’s Malaysian Adverse Drug Reaction Advisory Committee in 2013 confirmed that 11,437 cases of harmful drug reaction had been reported, and 0.2% were due to herbal remedies [1]. Thus, the evaluation of the toxicity of potential drug compounds has been enhanced in recent years [11]. Allopathy drugs and complementary and alternative medicines require a toxicology assay to identify any harmful effects that are not well-known until signs and symptoms appear after high consumption [12].

However, the main objective of toxicity research is to predict human toxicity via fast and accurate testing of many substances [13] based on model systems [14], and different routes of drug delivery systems such as using nanoparticles in the gastrointestinal tract [15] or chemical fertilizers and water retention as an example [16]. Previously, classical toxicity screening (including rodents, dogs, and rabbits) involved the compilation of data from a given laboratory for one substance at a time [17]; however, these tests are frequently expensive, time-consuming, and tedious [18]. This classic method focused on studying chemical compounds on phenotypic cell or animal results [19]. Thus, an attractive alternative approach involves applying the 3R concept of human-animal studies (i.e., reduction, replacement, and refinement), but it is not consistent with the use of rodent animal models [20]. Replacement: Zebrafish assays can be used to substitute such animal-toxicity studies using larval zebrafish, and it can be shown that larval zebrafish represent a critical system model; Reduction: zebrafish larvae can be used as a first-level model for toxicity to classify toxic drug candidates so that more stable drugs can be evaluated in mammalian models; Refinement: The model of embryonic and larval zebrafish provides a refined design to research organisms since embryos are partially fertilized and translucent in their early life [21]. It is recommended that the number of animals is limited, the test methods minimize pain and suffering of experimental animals, and approved substituted animal tests are used as much as possible [22].

The zebrafish (*Danio rerio*) is a suitable model for screening drugs for potential use to treat human diseases [23] based on phylogenetic analysis of fish and human genomes, which shows similar morphology and physiology of the nervous, cardiovascular, and digestive systems [18]. The zebrafish genome has been sequenced in full [24]. Zebrafish are a fast model for the study of genetic and de novo mutations [25]. The genes can inactivate in vivo, simulate human phenotypes, and obtain information on human diseases with genetic background through genomic editing approaches, such as CRISPR/Cas9 or artificial site-specific nucleases, zinc-finger nucleases, and transcription activator-like nucleases [25]. Zebrafish also provide a significant data file for vertebrate animals that enables researchers to anchor biochemical, genetic, and cellular hypotheses to high-performance observations at structural, functional, and behavioral levels [26]. The zebrafish embryotoxicity test, or fish embryotoxicity test (FET), is gaining popularity because it provides a total and well-defined developmental duration for a vertebrate embryo and allows the study of its early life stages [27]. As experimental models, both adults and zebrafish embryos are used [22]. For example, various compound tests (e.g., measurement of drug cardiotoxicity) have been conducted to assess drug effects in zebrafish efficiently [11]. Additionally, for the last two decades, zebrafish have been used to study angiogenesis, metastasis, anticancer drug screening, and an assessment of drug toxicity [28]. In summary, the cell structural and biochemical similarities between humans and zebrafish enable rapid forecasting of the possible impacts of chemical and other substances on human communities. Because zebrafish are becoming increasingly important as a test model, husbandry requirements to enhance the reproducibility and efficiency of this type of model in research environments are needed [29].

Toxicity studies generally begin in in vitro studies, with many different cell lines at different sample concentrations. The substance is then tested in several animal models, especially in mice and rats, before using on patients [30]. As mentioned in Table 1, there are several explanations why zebrafish as an animal model could be extremely useful for intermediate toxicity tests.

In vitro toxicity studies are inexpensive, fast, and easy, as shown in Table 1, but cultured cells are poorly associated with in vivo processes and therefore have limited translation benefit, while accurate data are obtained from laboratory rodent studies to extrapolate toxicants to humans [32]. Hence, the gold standard for predictive analysis of chemical risks to humans remains vertebrate toxicity studies; however, these studies are unsuitable for initial toxicity screening due to the reasons above and often involve significant amounts of a valuable test compound [26]. There has been a trend in the last 30 years towards the restricted use of higher animals in herbal toxicology studies in particular [22]. These animals have a series of restrictions. For example, rodents may be immune to the cardiotoxicity, especially when the endpoint is left ventricular contractile function. This may be due to rodents’ ability to substitute for myocyte failure by employing alternative mechanisms [25]. The zebrafish model is especially useful for the bioassay-guided identification of bioactive secondary metabolites [33]. The National Health Institute (NIH), USA, recently promoted a zebrafish model organism to study various genetically engineered diseases [34]. The Food and Drug Administration has recognized zebrafish tests for toxicity and safety evaluations for investigative newly developing drugs [26]. Large volume screening platforms exist in the form of a multi-wave plate format for testing chemical impacts on embryo development by assessing deformities, mortality, and structural characteristics across a range of concentrations [26]. Small molecules may be introduced directly into the water in multi-well racks, where fish take them up via diffusion. The researched drugs can also be injected into the yolk sac [35]. Thus, the zebrafish is a great model for studying the premature embryonic diet on toxicants [36]. Although researchers hope that both adults and embryos will prove useful for toxicology studies, embryos are ideal for toxicity examinations because of the transparency of the egg, which makes it easy to detect developmental phases and evaluate endpoints during the toxicity test [22,37]. This enables technical and economic benefits over rodent models [26,38] to reduce the number of substances and reduce the cost of animals in the development of drugs [39]. Up to 200–300 embryos can be developed per pair of adult zebrafish, while a typical pair of rodents only produce 5–10 descendants per matching occurrence [24]. Five days after post fertilization (dpf), the heart, liver, brain, pancreas, and other organs are created [40]. The three-lobe liver of zebrafish reflects humans’ biological function, including the absorption and production of lipids, vitamins, proteins, and carbohydrates [25]. One of the significant benefits of using zebrafish is that compounds can be easily supplied by adding water, similar to chemicals in the cell culture medium, which generally requires an overall amount of only 100 μL during growth [41]. Their optical clarity makes it easy to produce and recognize phenotypic properties during mutagenesis screening, and determine toxicity endpoints during toxicological analysis [32]. In addition, zebrafish embryos have been incredibly helpful in studying heart development and the functional effects of toxicants [14]. Figure 1 illustrates the temporal distinction between humans, rats, and zebrafish in early developmental life [26,41]. During development, there are still several limitations in the chemical identification of zebrafish. The fish is ectothermic and lacks heart septa, limbs, synovial joints, cancellous bones, lungs, and other organs [42]. Age is one of the most important limitations. Although chemical tests can be undertaken in adult fish, a high-performance scan using appropriate quantities of small molecules requires the animals to fit into a multi-well plate [19]. The analysis of gene expression throughout the larvae is feasible due to the optical transparency of the zebrafish tissues, which enables good penetration for light microscopy [32].

An alternative preliminary toxicity test is called the brine shrimp lethality test, which involves testing different toxicant concentrations. However, this technique does not reveal abnormalities or causes of death [43]. In contrast, using zebrafish as a screening model animal has some rational benefits, including large embryos, large numbers of embryos for testing, and simple visualization of organogenesis using fluorescence and transgenic strains [36]. Additionally, the ability to view biological processes provides the researcher with visual links to the organ system dynamics and, possibly, interorgan systems [44].

Today, many herbal products claim to provide pharmaceutical health benefits but do not provide any toxicological data [1]; thus, the safety of these drugs is questionable. This review aimed to compare the results of studies of the toxicity of medicinal plants and assess the toxicity tests used to analyze some of these commercial products from plants. This paper is the first review to assess the toxic and teratogenic effects of plant extracts on the zebrafish model to the best of our knowledge.

## 2. Zebrafish as a Suitable Alternative Animal Model for Toxicity Tests

One of the main goals of medicinal plant toxicology research is to identify many bioactive compounds with specific toxicity within a short period [13]. For many years, higher species have been used as models to measure medicinal products based on their toxicity [11]. For instance, recent studies of acute toxicity have mainly used mice as the animal model. Nevertheless, it is difficult to achieve a thorough and immediate toxicity check with this organism because of its strong state of breeding, high expenses, complicated procedures, and ethical restrictions [45]. Instead of using rodents, fish are the best candidate for this purpose. Fish traditionally have been used in toxicity testing of individual substances and effluents [46], and today zebrafish are commonly utilized to test for developmental toxicity, general toxicity and to carry out medication screening as a credible vertebrate model [11,26]. Hence, the first large-scale studies for bioactive molecules using zebrafish embryos were published, using the merits of zebrafish as phenotypic testing models to evaluate the effects of biomolecules and explore different bioactive compounds [47]. The zebrafish is widely used in numerous subfields of biology, and without a doubt, it is one of the leading species in various research areas, including developmental biology, ecotoxicology, and genetics [48]. Zebrafish embryos can also quickly consume tiny molecular compounds, thus providing a valuable model for drug testing and evaluating teratogenic effects [49,50] of exposure to toxic compounds [38]. Zebrafish embryos and larvae are outstanding models for testing the toxicity of substances, especially if those substances are present in low quantities [51]. Zebrafish larvae are very useful in imaging studies, and they likely will prove useful for non-imaging endpoints. Scientists are discovering new innovative paths for evaluating biochemical processes [19]. Within 5–6 days dpf, zebrafish development reflects the full developmental period of a vertebrate embryo before it becomes self-sustainable and, therefore, this substitute organism is not currently recognized as a genuine in vivo form by European law [52].

Teratology is the study of unusual growth, and a teratogen is any substance that triggers the production of a congenital anomaly or enhances the occurrence of a specific hereditary deficiency [53]. Screening for teratogenicity includes introducing zebrafish embryos to the required concentration of the compound of interest, and it has become a popular model for analyzing the teratogenic effects of medicines [35]. Teratological and embryo-toxic effects are easy to detect due to the transparency of zebrafish embryos during their growth outside of their parent [54]. Teratogenic effects include tail malformation, pericardial edema, malformation of the notochord, scoliosis, yolk edema, and growth delays [22]. Therefore, it is essential to evaluate the embryotoxic and teratogenic toxicity of medicinal plants on embryo growth. Embryo malformations may be caused by activation of the Caspase-3 enzyme, which is the leading cause of apoptosis [55], or by other factors such as reactive oxygen species-induced oxidative stress [6].

The effectiveness of the zebrafish model system was evident in the 1960s and 1970s, as numerous studies used zebrafish as bioassays of chemicals affecting normal functioning and reproductive success. During the 1980s and 1990s, many researchers verified that zebrafish are a convenient and reliable option for comprehensive toxicological screening and early life and lifetime exposure tests [56]. In 2000, the first chemical screening procedure was reported that allowed researchers to test a very slight concentration of compounds on live zebrafish in 96-well plates [19]. To date, there are ten types of tests used to study toxicity in zebrafish: (1) the zebrafish embryo toxicity test (FET); carcinogenicity; (2) developmental toxicity and teratogenicity assessments; (3) reproductive toxicity assessment; (4) behavioral toxicity assessment; (5) endocrine disorders; (6) acute toxicity; (7) neurotoxicity; (8) optical (ocular toxicity); (9) cardiotoxicity; and (10) vascular toxicity [31]. Among the available tests, FET is the most useful for assessing chemical and substance toxicity in zebrafish [48]. Numerous toxicity studies of specific chemicals have shown a significant correlation between results for zebrafish embryos and acute toxicity of fish [57]. The FET test aims to determine the acute toxicity of the embryonic phases and establish variables for zebrafish [52]. It is based on mortality and teratogenesis of the zebrafish embryos [46]. FET is a valuable option for replacing the use of adult animals to evaluate toxicity and enhance existing toxicity assays [48]. Advantages of this test include:A wide range of chemicals may be relevant;Embryos are short-lived, susceptible, cost-effective, and have low variability;The regulatory and scientific communities consider them to be standardized;The tolerance of various species or organisms can be compared [58].

The possibility of tracking many toxicity endpoints, including alteration of molecular processes and malformations, is one of FET’s main advantages with *Danio rerio* [59]. With the FET test, acute toxicity is assessed based on positive results, and embryo formation is examined every day for any unusual developmental phenotypes, especially morphological defects, including (i) coagulation of fertilized eggs; (ii) lack of somite formation; (iii) lack of detachment of the tail-bud from the yolk sac; and (iv) lack of a heartbeat [52]. Nagel (2002) outlined these toxicity indicators as measured using an inverse light microscope based on “yes” or “no” responses to the presence of the four factors [60]. On the other hand, Hermsen et al. [61] introduced a new model of embryotoxicity (ZET) using the general morphology score (GMS). A separate scoring list was established for teratogenic impact, which helped track slow progress, developmental delay, and teratogenicity. However, the ZET, along with GMS, operates as an essential and useful test method for screening the chemicals’ embryotoxic properties in the compound groups studied [27]. Furthermore, the MolDarT was created with zebrafish eggs/larvae to develop a molecular mechanism test method [61]. It is based in theory on the DarT (*Danio rerio* teratogenicity test), established by Nagel et al. [60], which reveals and tracks the developmental impact and deformities of freshly fertilized zebrafish eggs within 48 h [62].

Experiments to test toxicity include acute, sub-chronic, and chronic studies of compounds’ toxicity to specific organ pathways and hypothesis-driven research [31]. Zebrafish were found to be a useful tool for the comprehension not only of neurotoxicants’ structural and chemical effects but also for the assessment of behavioral dysfunction correlated with such toxicity [38]. Several behavioral endpoints are used in neurotoxicity studies to determine the therapeutic influences of medications and their neurotoxins on neuron development in zebrafish [63]. There is an incomparable system to identify endocrine activity lacking significant morphological abnormalities [13]. For the reasons described above, zebrafish toxicity screening provides many significant experimental advantages, including embryo and larvae visibility, high-efficiency short test time, low number of necessary compounds, ease of handling, and direct delivery compounds [54]. The zebrafish embryo develops quickly outside the mother’s body and enters maturity in a couple of months, among other benefits [38]. Additionally, more than 70% of the disease-associated genes are similar to those found in human diseases [50]. Furthermore, the physical, biochemical, genetic, and molecular makeup of zebrafish, including organs and tissues, has been demonstrated to be identical to their mammalian relatives. Metabolites, signaling mechanisms, and neurological and cognitive structures are similar to those in mammals [54].

A compound’s toxicity is measured using two indicators: the median effective concentration (EC_50_) and the median lethal concentration (LC_50_). The concentration-response curve is generated using 24 h of data to obtain the EC_50_ (teratogenic effects) and LC_50_ (embryotoxic effects). The ratio of LC_50_ to EC_50_ is the corresponding therapeutic index (TI). In pharmaceutical administration, a TI value < 1 is optimal [50,64]. The strength of the association between zebrafish embryo LC_50_ values and rodent LD_50_ values for 60 different compounds was demonstrated in one particular laboratory study by Ali and colleagues [42]. Test protocols for the use of zebrafish embryos (FET) in the Organization for Economic Cooperation and Development (OECD) test guideline 236 were tested and applied [65]. As determined by OECD guidelines, the value of LC_50_ is calculated based on lethality, coagulation, lack of somite formation, heartbeat failure, and a lack of detachment of the tail. In contrast, teratogenic effects are used to compute the EC_50_ values [22]. Parng et al. (2002) showed that the toxicity of the drug tested (ethanol) was similar for zebrafish and mammals with log LC_50_ values of 4.0 and 3.8 mg/mL, respectively [66]. Throughout the screening steps, zebrafish bioassays have been used to classify the active fractions based on endpoints developed and tested in other species, which led to identification of the actual effect of extracts on different life stages in zebrafish (Table 2).

Additionally, indicators such as delayed growth, restricted movement, irregular head-trunk angle, scoliosis/flexure, and yolk sac edema have been analyzed in developing zebrafish [73]. However, recent findings indicate that herbal products’ metabolism using zebrafish can represent actual outcomes of mammalian methods, while single in vitro methods cannot do so [39]. Table 3 lists some toxicity screening studies of medicinal plants that used zebrafish as the model organism.

## 3. The Most Relevant Behavioral Effects and Basic Methodology Based on Toxicological Assessment Observed in Zebrafish Larvae

Previously, zebrafish were used for toxicity evaluations of agrochemicals, but recently they have been used for toxicity assessments for therapeutic compounds [11]. Of note, the zebrafish response is also a sensitive predictor of irregular toxicity changes [34]. In particular, specific tasks have been created or changed to test the behavior of zebrafish compared to rodent models. Quite basic swimming steps and the ability to capture/consume can often be helpful [38]. The creation of behavioral tests that can employ repeated tests in a short window (i.e., within 24 h) may prove to be the perfect compromise, and some researchers have begun designing tasks that use multiple tests to measure their acquisition during a single session [38]. If one particular mode of action is to be detected, various experimental parameters can contribute to incoherent actions (hypo-or hyperactivity) responses of zebrafish when determining toxicity [122]. As a consequence, several behavioral evaluation methods have been developed to address different behavioral endpoints, including spontaneous tail coiling, photomotor response (PMR), locomotor response (LMR), and the alternating light/dark-induced locomotor response (LMR-L/D) [122], which we will not discuss further in this review.

Zebrafish are easy to control in toxicology screening tests using different platforms (exposure format) for detecting parameters. The type of exposure format is classified into ten categories, six of which are useful. Zebrafish body length at 5 dpf is about 4 mm, so a multiple-well plate can be used to incubate zebrafish larvae [17]. However, embryos cover a wide range of sizes based on the performing test, so 6-well to 96-well transparent plates can be used. Analytical experiments that use microplates, such as cell-based evaluation, can be conducted and applied to high-performance primary drug screening [123]. Furthermore, zebrafish larvae fit onto 96 or 386-well microplates [24]. Because the zebrafish embryo is tiny, can be handled easily [31], and requires only a small number of compounds per test, they can be screened in a 96-well microplate format [123]. However, OECD TG236 suggests using a 24-well plate (with a medium of 2 mL per well), as it overcomes the problem of high concentration developing in conjunction with evaporation [17]. Nevertheless, TG236 specifically checks for lethality and only considers developmental defects indirectly [17]. Table 4 lists several studies that analyzed embryotoxicity and apoptotic induction.

Apoptotic functions are identical in zebrafish and humans [66], and can be easily detected using fluorescent labeling methods [123]. The zebrafish model was used to discover the toxicity of 69 plants in this review, including 88 crude plant extracts, eight polyherbal formulates/commercial products, and two phytocompounds. In terms of exposure format, out of 58 studies described in Table 4, 25 used 24-well plates, and 10 used 96-well plates.

## 4. Using Zebrafish Embryos to Detect Developmental Toxicity

It seems that the number of zebrafish screening tests to evaluate the toxicity of compounds will keep increasing each year [19]. The use of fish embryos was proposed as an alternative screening method to assess fish’s acute toxicity [21,57]. In transparent species, the effects of compounds on different organs, such as the brain, heart, cartilage, liver, intestine, and kidney, were identified without arduous screening [66]. This specific trait also makes it feasible to quickly measure toxicity endpoints of multiple substances [6,124], highlighting the efficacy of toxicity models using zebrafish embryos [66]. For example, after 2005, regular sewage surveillance monitoring of fish embryotoxicity has been made compulsory, substituting conventional fish tests no longer approved for standard whole effluent assessment [62].

From the egg phase, zebrafish embryos can survive by ingesting yolk and can visibly be tested for malformation for a few days in a single well of a microplate [32]. When the embryo is not clumped, has a cardiac beat, has fully shaped body sections, and the tail is detached from the yolk, the standard endpoints of embryogenesis have been said to have occurred [28]. The main characteristics of embryogenesis and the genetic basis of growth in zebrafish have been widely researched [125]. Another significant indicator for assessing toxicity is the hatching rate, and zebrafish embryos start to hatch at about 48 h post-fertilization (hpf) under ordinary conditions [8]. Delayed growth of zebrafish embryos can result in a low hatchability and can thus be among the critical aspects of the sub-lethal effects of plant extracts [53].

The essential cell structure and biochemical similarities between humans and animals enable researchers to use the zebrafish model to forecast the possible impacts of chemicals and other human communities [10]. Furthermore, it is easier to study the biochemical mechanisms of significant zebrafish organs, whereas histological analysis required in the mouse model is more difficult [66]. Testing parameters commonly used for zebrafish include embryo survival rate, lethality, behavior, and organ deformity; researchers have found that zebrafish exhibit a strong dose reaction to toxicity, making it a useful animal model for toxicity screening [123]. Previously, we mentioned FET; most studies use OECD guidelines to measure toxicity of crude extract/fractions, and those guidelines for FET set levels as follows: dangerous (10 mg/L < LC_50_ < 100 mg/L), toxic (1 mg/L < LC_50_ < 10 mg/L), and carcinogenic (LC_50_ < 1 mg/L) [126].

Zebrafish developmental toxicity testing is a type of FET (FET can be used for any fish species), and it is primarily aimed at supplementing developmental toxicity screening in mammals [17]. When detecting developmental toxicity in zebrafish embryos, endpoint parameters are categorized in two ways compared to a standard (negative control). First, lethality is assessed based on coagulation, malformation of somites, absence of heartbeat, and lack of tail detachment. Second, teratogenicity is evaluated based on abnormal eye development, lack of spontaneous movement, unusual heart rate, lack of pigmentation, and edema. The standard shows normal development [60,127].

## 5. Defects in Zebrafish Organs Found in Developmental Toxicity Studies

In developmental toxicity studies, the mortality of zebrafish embryos occurs before 24 hpf or just after hatching, and cardiac deformities are sometimes detected, such as pericardial edema, abnormal heart form due to edema or aplasia, and irregular heartbeat [4]. The motility of the zebrafish embryo can also be affected by toxins, as exhibited by coagulation, heart failure, and non-development of the yolk embryo’s tail [112]. Other factors, such as reactive oxygen species-induced oxidative stress, are also assumed to be linked to abnormal development during embryogenesis [128]. A teratogenicity test is carried out to assess developmental toxicity (i.e., if the compound has teratogenic effects on the embryos). After treatment, the development stages of embryos are observed under an inverted microscope to look for malformations based on the numerical system designed from 5 to 0.5, where 1 indicates severe malformation, 4 indicates slight malformation and 5 is totally normal [50]. The teratogenic score is calculated as follows:(1)$$\mathrm{Teratogenicity}\;\%\;={\left[\frac{\mathrm{Malformation}\;\mathrm{score}\;\mathrm{at}\;\mathrm{each}\;\mathrm{developmental}\;\mathrm{stage}}{Total\;\mathrm{score}}\right]}^{}\times \;100$$

In such analyses, it is essential to specify the observational setup and endpoint evaluation (i.e., which particular behavioral change(s) will be analyzed). Below we discuss the defects to vital organs of zebrafish that can occur during the toxicity assay.

### 5.1. Heart (Cardiotoxicity)

The zebrafish is an excellent model organism for studying organogenesis and the cardiovascular system [86,129]; thus, it was developed quickly as a cardiovascular research model organism [35]. The first active organ that develops in zebrafish is the heart [73,130,131]. The heartbeat is a primary and relevant sub-lethal endpoint in the embryonic fish toxicology assay [73,130]. It is regularly assessed as an indicator of toxicity in zebrafish embryos [73] and estimates fish metabolic function as a biological factor [6,132]. In zebrafish, the normal embryonic heartbeat of 120–180 beats per minute is close to that of the human heartbeat. At the embryo pharyngula level, the heartbeat pulse can be tracked because the tail is visibly pigmented [102,118]. Ismail et al. reported that the most vulnerable stage for external stimulation (e.g., chemicals, toxicants, and physical pressure) is the early growth stage of embryos [2]. In toxic environments, pericardial edema can be caused by the general response. Pericardial edema in zebrafish embryos can be caused by many different toxicants [133], and changes in heartbeat rate may be a common reaction to toxicant exposure [130]. The toxicity mechanism of a compound is due to activation of Na^+^ and K^+^ inhibitors, and these Na^+^ channels likely trigger the heart rate. An elevated heart rate can overwhelm the myocardium and cause operational and biological damage to the heart of zebrafish [134]. For example, studies of the effects of medicinal plants such as *Andrographis paniculata*, *Cinnamon zeylanicum*, *Curcuma xanthorrhiza*, *Eugenia polyantha*, *Orthosiphon stamineus* [2], *Tinospora cordifolia* [118], *Carthamus tinctorius* [83], and *Euodia retaecara* [92] on zebrafish embryos reported a decreased heartbeat. In contrast, embryos treated with curcumin showed a heartbeat increase [1].

The heart is anteroventral in zebrafish and lies between the operculum and pectoral girdle in the thoracic cavity [35,135]. It is divided into the atrium, sinus venous, ventricle, and bulbous arteriosus, it starts beating at about 22 to 26 hpf [17,136], and produces a full set of ion channels and metabolic regulation [123]. In later stages, the normal function of the heart plays a leading role in the growth of the embryo. An abnormally formed cardiovascular system can result in the unusual overall growth of the animal, severe malformations, and a malfunctioning body [8]. Additionally, when the heart is swollen due to the active compounds in the pericardial sac, cardiac cells become irritated [43]. The following method is used to measure a normalized cardiac rate [120]:(2)$$\%\;\mathrm{Normalized}\;\mathrm{cardiac}\;\mathrm{rate}\;=\;\frac{Heart\;rate\;\left(sample\;test\right)}{\mathrm{Heart}\;\mathrm{rate}\;\left(\mathrm{vehicle}\right)}\;\times \;100$$

Zebrafish embryos are also being used in cardiotoxicity (reduction of heartbeat) studies to model various human diseases [11]. The morbidity and mortality of people with cancer are associated with cardiotoxicity [35]. Most common drugs have cardiotoxic effects and affect the heart in zebrafish model systems [86]. A very accurate zebrafish cardiotoxicity assay assessing potential drug toxicity to the human cardiovascular system has been documented [11]. Thus, the effects of cardiotoxicity, such as minimal or no blood flow, may be explained by the fact that a large percentage of apoptotic heart cells lead to underdevelopment of the heart and pericardium, which in turn can cause an unusual heartbeat and a delay in body development (growth retardation) [137,138]. Gao et al. reported that the expression of two heart markers (*amhc* and *vmhc*) reflected the seriousness of heart failure in zebrafish [133].

At day 3 of the formation of zebrafish embryos, the primary and sprout vessels become active, and the blood vessel structure is quite like that in the human body [91]. A single endothelial cell layer surrounded by supporting cells forms the vessels that constitute the circulatory system. Endothelial precursor cells result from vasculogenesis and angiogenesis within the lateral mesoderm of mammal embryos. Vasculogenesis explains the de novo establishment of blood vessels by angioblast coalescence. In comparison, angiogenesis includes the creation of new vessels by earlier developed vessels and is typically described by the sprouting of an endothelial cell [139]. Kinna et al. reported the potential role of zebrafish *crim1* in regulating vascular and somatic development [140].

### 5.2. Gills

Gills are essential to fish because they are the leading site for gas exchange [141,142,143], and they engage in osmoregulation [90], ionic control, acid-base balance, and excretion of nitrogen waste in fish [144]. Several studies on the histological organization and physiology of gills in many fishes have been carried out [145]. Enhanced green fluorescent protein (EFGP) is a successful marker for toxic chemicals in transgenic zebrafish. One of the benefits of this zebrafish transgenic model system is that spatio-temporal patterns of EGFP expression can be assessed during initial stages after exposure to a particular toxicant [146]. A study by Seok and colleagues found that the most effective and earliest expression of EGFP was seen in zebrafish gills, indicating that the gill is the most sensitive tissue and is impacted at relatively low metal concentrations [146]. The studies conducted by Rajini et al. [147] showed that visible damage such as inflammatory cell infiltration in gills, minimum aggregation in primary lamellas, secondary lamellae fusion, diffuse epithelial hyperplasia, and multifocal cell mucosal hyperplasia occurred in zebrafish exposed to sublethal concentrations of a combination of pesticides [147]. Generally, in zebrafish, gills exposed to toxicants from 48 to 72 h show fusion, and clubbing of distal lamellar regions [148]. However, for herbal toxicity, this organ has not fully been considered.

### 5.3. Tail

One of the endpoints of embryonic growth in zebrafish is when the tail is detached entirely from the yolk [28], which can be affected by toxicants. The tail emerges during the larval stages from the primordium in the ventral fin fold, which coincides with a gap in the melanophore line immediately before the back tip of the notochord [149]. Tail kink and tail bending are malformations caused by toxicity. However, tail malformations and spinal axis disabilities can also be affected by predation and a drastic decrease in the food supply [6].

### 5.4. Yolk Sac

Like human embryos, zebrafish embryos possess a protruding yolk sac, but in zebrafish, it serves as a nutritional reservoir for the embryo [36,43]. Yolk sac edema is a common pathology in toxicity tests of zebrafish [32,150], and it is a sign of reduced nutrient adsorption by the embryo [43]. It may be affected by overhydration of osmoregulation and toxin accumulation in the yolk sac [150]. Malformation in other organ systems that help allocate nutrients may be the trigger for this deficiency. This deficit could also contribute to nutritional absorption, undernourished embryos, and embryo mortality [43]. After maternal exposure, toxicants can be placed in the yolk sac as well [36]. As they are deposited in the yolk, the embryo becomes directly exposed to these toxicants, which allows for definitive exposure timing. Hence, a finite quantity of maternally stored embryo yolk is used in the zebrafish embryo for early feeding, and it provides an understanding of embryo nutrition processes and disruptions that occur due to toxicity exposure [36].

In zebrafish developmental toxicology research, there are numerous classes of yolk phenotypes. Yolk sac edema is quite like yolk edema. Yolk retention refers to decreased mobility/use or malabsorption of the yolk. If the area of the yolk sac is significantly greater in a test fish than in the control fish, it suggests that yolk absorption has been affected. On the other hand, the rapid use of yolk leads to a smaller yolk area and indicates enhanced movement or use of the yolk [36].

### 5.5. Hatchability

A zebrafish embryo is assumed to have been hatched once it is entirely out of the chorion. Certain drugs and extracts can affect hatchability. Lack of hatchability may imply a growth lag or slow development [102]. For example, zebrafish embryos exposed to *Millettia pachycarpa* extract exhibited a considerable dose-dependent reduction in embryo hatching [6]. Zebrafish embryos start to hatch at 48 hpf in ordinary conditions [8], but percent hatchability after 72 h post-treatment exposure (hpte) is calculated using the following formula [73]:(3)$$\%\;\mathrm{Hatchability}\;=\;\frac{\mathrm{No}.\;\mathrm{of}\;\mathrm{hatched}\;\mathrm{embryos}}{\mathrm{Initial}\;\mathrm{no}.\;\mathrm{of}\;\mathrm{embryos}}\;\times \;100$$

According to the literature, this model to determine the toxicity of crude extracts or bioactive compounds, and toxicological effects can be compared.

## 6. Chemical Studies of Embryotoxicity

Changes in fish behavior are sensitive indicators of unintended chemical pollution [151]. The zebrafish model is vital in evaluating the effects on embryonic/larval activity of developmental exposure to harmful compounds [38]. Comprehension and monitoring of these considerations and future revision/harmonization of procedures will reduce the uncertainty of the outcomes for chemical hazard assessments [122].

### 6.1. Salinity

Several early life researches have been carried on the effects of salinity on zebrafish [152]. When larval zebrafish are subjected to mild environmental stressors, severe salinity shifts lead to a quick locomotive reaction [153]. Lee and colleagues showed that larvae (5 dpf) treated with NaCl exhibit dramatically increased locomotive activity and concentration-dependent reactions. A linear relationship between whole-body cortisol and sodium chloride in the concentration range of 5.84 ppt was shown in zebrafish larvae [154].

### 6.2. Phytochemical Compounds

Yumnamcha et al. [6] reported the presence of saponins, alkaloids, phenolic compounds, and triterpenoids in the aqueous extract of *Millettia pachycarpa*. Potent hemolytic activity was found to lead to acute cell membrane death in zebrafish. The toxic function of saponin in the zebrafish embryo is uncertain. However, prior studies have shown that it can be caused by physicochemical characteristics (surfactant) and/or membranolytic impacts on the chorion, a semipermeable membrane that covers the embryo until it hatches [6,155]. Moreover, according to Liu et al. [156] and Yumnamcha et al. [10], some bioactive compounds such as saponin mixtures and alkaloids can cause damage to DNA. *Artemisia capillaris* was evaluated for its toxicity based on embryotoxicity, and isofraxidine 7-O-(6′-O-p-coumaroyl)-β-glucopyranoside was isolated from this plant. The toxicity, death, and cardiac rates of handled zebrafish embryos are used to diagnose safe and effective concentrations [157].

In contrast to flavone, artificial flavonoids such as kaempherol, 7-hydroxyflavone, 6-methoxyflavon, and 7-methoxyflanone induced considerable toxicity in zebrafish larvae [158]. Additionally, the lack of flavonoids can potentially trigger oxidative stress to cellular molecules, which leads to lower survival rates of zebrafish embryos [12]. Bugel et al. [63] reported that 15 of 24 flavonoids affected at least one or more behavioral and developmental endpoints in zebrafish. However, they focused on two endpoints in their experiment: abnormal spastic behaviors (72 hpf) and alternations in the larval photomotor responses assay (120 hpf). Li et al. [99] exposed zebrafish to Xiaoping at high doses, and many larvae opted to swim at a medium to a low rate, which led to decreased locomotion ability. Changes in larval behavior have been confirmed to be associated with damage to the central nervous system. The function of neural population numbers in animals with a broader central nervous system is being measured using fluorescence and new devices developed over the last decade [159]. In one study, zebrafish embryos subjected to different caffeine dosages demonstrated a decrease in susceptibility to touch-induced movement. Such abnormalities were linked to changes in muscle fibers and axon projections of both primary and secondary motor neurons through immunohistochemistry [160]. Furthermore, toxicological testing may also impact other environmental variables (temperature, water, pH, total hardness or dissolved oxygen) [161].

## 7. Conclusions

The data collected in this study suggest that zebrafish embryotoxicity tests are able to evaluate drug toxicity and that the zebrafish model offers a suitable replacement for laboratory animals such as rats, mice, and rabbits. As a toxicology model, zebrafish can expose developmental toxicity mechanisms because they are close to mammals. Zebrafish embryos and larvae showed significantly higher susceptibility to toxins than did adult zebrafish. In this review, most of the extracts were polar, such as ethanol, methanol and aqueous extracts, which were used to detect the toxicity and bioactivity. However, the use of the zebrafish model will provide insight into the mechanisms of toxicity of medicinal plants and will help identify and discover new medications for the treatment of human diseases. The zebrafish model is planned as a replacement for models based on higher vertebrate animals to study medicinal plants’ toxicity.

## Figures and Tables

**Figure 1 plants-09-01345-f001:**
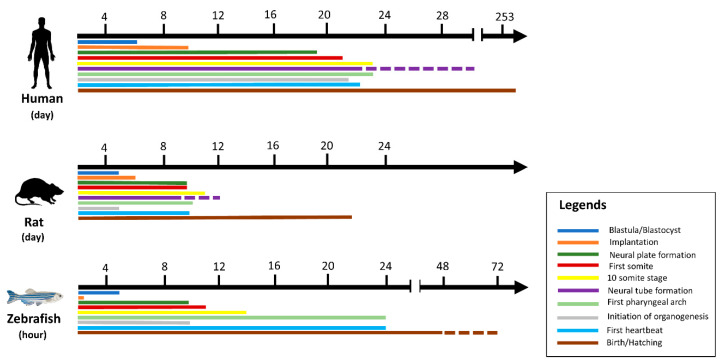
Comparative analysis of human, rat, and zebrafish based on the early stages of development.

**Table 1 plants-09-01345-t001:** Suggested methods for toxicological analysis [30,31]

	Cells	Fruit Fly	Zebrafish	Rodents	Humans
**Benefits**	Fast, simple, inexpensive	Short period of generation	Fast, simple, inexpensive	Complexity of body	High value translation
	Well-standardized	Work easily with	Complexity of body	Adequate predictiveness	Credible and logical
	Many probable readings	Low maintenance costs	Ethical examination in vivo	Administration route	Dosage details
**Limitations**	Modest predictivity	Genetically far from human beings	Modest flexibility	Time-consuming and costly	Time-consuming and costly
	Simplified process	Simple anatomy	Moderate foresight	Comprehensive regulation	Comprehensive regulation
	Low value translation	No efficient immune system	Moderate value translation	Ethical limits	Ethical limits

**Table 2 plants-09-01345-t002:** Zebrafish bioassays and measured endpoints.

Life Stage	Measured Endpoints	References
Egg	Survival, hatching, teratology	[1,60,67]
Larvae	Survival, abnormalities, enzyme activities and behavior	[67,68,69,70]
Adults	Survival, behavior, enzyme activities and genotoxicity	[71,72]

**Table 3 plants-09-01345-t003:** Existing studies that screened toxicity of medicinal plants using the zebrafish model.

Plant/Family	Common Name	Extraction/Part of Plant	Concentrations Range (µg/mL)	LC_50_ (µg/mL)	Characteristics of Effect	Reference
*Acorus calamus*/Acoraceae	Sweet flag	Ethanol/rhizome	n.a.	n.a.	Fish treated with this extract returned to normal condition and revealed a decrease level in superoxide dismutase and decreased mitochondrial volume and cell viability.	[74]
*Achyranthes aspera*/Amaranthaceae	Chaff-flower, prickly chaff flower, devil’s horsewhip, burweed	Ethanol/leaf	n.a.	n.a.	Fish treated with this extract returned to normal condition and revealed decrease in superoxide dismutase level and decreased mitochondrial and cell viability.	[74]
*Achyranthes bidentate*/Amaranthaceae	Ox knee, niu xi (Chinese)	Aqueous/root	0–30.01 µg/mL	No mortality observed	The extract did not exhibit noticeable lethal or severe side effects on zebrafish larvae/embryos.	[75]
*Allium carinatum*/Amaryllidaceae	Keeled garlic, witch’s garlic	70% aqueous methanol/whole plant	1–60 µg/mL	55.8 µg/mL	The extract reduced developmental toxicity and neutropenia. No teratogenic effects were seen.	[76]
*Allium flavum*/Amaryllidaceae	The small yellow onion, yellow-flowered garlic	70% aqueous methanol/whole plant	1–60 µg/mL	50.3 µg/mL	The extract reduced developmental toxicity and neutropenia. No teratogenic effects were seen.	[76]
*Andrographis paniculata*/Acanthaceae	Green chiretta, creat	Aqueous/leaf	0–10,000 µg/mL	525.5 µg/mL (48 hpf) 525.6 µg/mL (96 hpf)	Morphological defects were observed at 96 hpf after exposure to teratogen concentration.	[2]
*Annona squamosa*/Annonaceae	Sugar apple, sweetsop	Ethanol/young leaf	100–1000 µg/mL	n.a.	Late hatching process and morphological deformations occurred at higher concentrations up to 800 µg.	[77]
*Avicennia marina*/Acanthaceae	Grey mangrove or white mangrove	Methanol/bark, leaf, stem, flower and fruit	50–100 µg/mL	n.a.	n.a.	[78]
*Azadirachta indica*/Meliaceae	Neem	Ethanol (cold)/fruit	1000–5000 µg/mL	n.a.	Not toxic to adult zebrafish and did not alter the locomotor system.	[79]
*Bougainvillea galbra*/Nyctaginaceae	The lesser bougainvillea, paperflower	Aqueous/bracts	1–300 µg/mL	85.51 µg/mL	Yolk sac edema was observed at different concentrations. Hypopigmentation in the embryo was caused by the purple bract extract at 30 µg/mL. Overall, this extract generally showed moderate embryo toxicity.	[80]
*Bixa Orellana*/Bixaceae	Achiote	Hot water/leaf	50–10,000 µg/mL	n.a.	Plant extract affected the hatchability of embryos and was linked to delayed development. Also, a coagulated embryo, and tail malformation were observed as teratogenic effects.	[81]
*Capsicum chinense*/Solanaceae	Habanero type pepper	Ethanol/fruit	0.39–100 µg/mL	39.7 ± 2.1 µg/mL	The embryo demonstrated late development represented by an absence of pigmentation of the tail, and irregular formation of the somites demonstrated the noticeable tail curve with a wide end.	[82]
*Carthamus tinctorius*/Asteraceae	Safflower (English), Kashefeh (Persian)	Aqueous/flower	40–1000 µg/mL	345,600 µg/L	Hatching inhibition, depressed heart rate, abnormal spontaneous movement, pericardial edema, yolk sac edema, unusual head-trunk tilt, suppression of melanin release, enlarged yolk, and short body length were identified.	[83]
*Clinacanthus nutans*/Acanthaceae	Sabah snake grass	Hexane/leaf	15.63–500 µg/mL	75.49 µg/mL	Morphological disorders (less pigmentation, tail bending, edema, abnormal yolk sac).	[84]
*Cinnamon zeylanicum*/Lauraceae	True cinnamon tree, Ceylon cinnamon	Aqueous/bark	0–10,000 µg/mL	985.8 µg/mL (48 hpf) 50.58 µg/mL (96 hpf)	This plant showed minimum embryotoxicity on zebrafish. In 72 hpf embryos, higher concentration of extract provoked yolk sac and pericardial edema.	[2]
*Croton tiglium*/Euphorbiaceae	Purging croton, Jamaal gota (Hindi)	Aqueous/seed	4000–24,000 µg/mL	11,880 µg/mL (24 h) 8160 µg/mL (48 h)	Signs of stress, enhanced respiratory rate, jerky movements, loss of equilibrium, circular swimming before losing balance.	[10]
*Curcuma longa*/Zingiberaceae	Turmeric	Methanol/rhizome	7.8–125 µg/mL	92.42 µg/mL (24 hpf) 79.20 µg/mL (48 hpf) 68.32 µg/mL (72 hpf) 56.68 µg/mL (96 hpf) 55.90 µg/mL (120 hpf)	Physical anatomy deformities as teratogenic effects; tail bending, yolk sac edema extension, and a curved trunk were observed at high concentration among hatched embryos.	[1]
*Curcuma longa*/Zingiberaceae	Turmeric	Methanol/rhizome	No concentration ranges were mentioned (10 µg/mL used)	n.a.	No visible symptoms of toxicity were observed: no change in heart rate, body impairment, absence or delay in reaction to tactile stimuli, or mortality.	[85]
*Curcuma xanthorrhiza*/Zingiberaceae	Javanese ginger, Temulawak	Ethanol/rhizome	100–500 µg/mL	180.52 µg/mL at 48 h 79.55 µg/mL at 96 h	Major malformations of the pericardial edema of embryos at a concentration of 100 µg/mL, yolk sac edema and tail malformation also were observed.	[43]
	Javanese ginger, Temulawak	Aqueous/rhizome	0–10,000 µg/mL	748.6 µg/mL (48 hpf) 703.7 µg/mL (96 hpf)	Decreased survival rate, organ deformity, unusual heartbeat, and delayed hatching rates.	[2]
*Cynodon dactylon*/Poaceae	Scutch grass	Hexane, chloroform, acetone, methanol/n.a.	10–100 µg/mL	32.6 µg/mL	The period of systole and diastole was decreased by the methanol extract.	[86]
*Derris elliptica*/Fabaceae	Tuba root (Indonesia), opay (Philippines)	Aqueous/leaf	50 (0.05%), 500 (0.5%) µL/mL	n.a.	The concentration of 0.5% led to the early death of embryos: unformed head, unformed tail, coagulation, and death. A delay in growth and restricted movement compared to the control group were present at 0.05%.	[53]
*Dieffenbachia amoena*/Araceae	Spotted dumbcane	Aqueous/leaf	0–10,000 µg/mL	1190 µg/mL	Growth retardation, no heartbeat was noticed at 500 µg/mL and higher concentrations (cardiotoxicity).	[87]
*Diospyros discolor*/Ebanaceae	Velvet persimmon, velvet apple, mabolo tree	Aqueous/leaf	0.05–10%	1%	Tail malformation, delayed growth, head abnormality, yolk malformations, and abdominal swelling were observed.	[88]
*Eclipta prostrata*/Asteraceae	False daisy, yerba de tago, Karisalankanni and bhringraj	Aqueous/leaf	0.01–40%	1%	At the concentration of 0.01% v/v and 1% v/v, did not observe any mortality and abnormality	[89]
*Enydra fluctuans*/Asteraceae	Marsh herb, water cress	Ethanol/leaf	12,500–400,000 µg/mL	24 h: 204,132 µg/L 48 h: 170,513 µg/L 72 h: 139,478 µg/L 96 h: 92,956 µg/L	The death rate was found as the exposure period was extended from 24 to 96 h, the median lethal concentration decreased. A negative relationship between exposure time and LC_50_ was found.	[90]
*Eugenia polyantha*/Myrthaceae	Bay leaf	Aqueous/leaf	0–10,000 µg/mL	921.2 µg/mL (48 hpf) 60.39 µg/mL (96 hpf)	Decreased survival rate, organ deformity, unusual heartbeat, and delayed hatching rates.	[2]
*Euphorbia pekinensis Rupr*/Euphorbiaceae	The Peking spurge, daji (Chinese)	Aqueous/commercial raw herb	100–250 µg/mL	250–300 µg/mL (previous study)	n.a.	[91]
*Euodia rutaecarpa (Tetradium ruticarpum)*/Rutaceae	Goshuyu (Japanese), Bee tree	Evodiamine (Commercial product)	0.05–1.6 µg/mL	0.354 µg/mL (LC_10_)	Evodiamine (bioactive compound) had a 10% lethality at 354 ng/mL and caused heart defect, changes in heartbeat and blood flow, and pericardial malformation. Also, it could cause cardiovascular side effects involving oxidative stress.	[92]
*Excoecaria agallocha*/Euphorbiaceae	Back mangrove	Methanol/bark, leaf and stem	50–100 µg/mL	n.a.	Higher concentrations induced intense impacts on embryo melanogenesis. Those concentrations also reduced the eye melanin material of the embryo.	[78]
*Ficus glomerate*/Moraceae	Cluster fig tree, Indian fig tree, goolar fig	Hot aqueous/leaf	125–2000 µg/mL	239.88 µg/mL	Lower hatchability, body size, heartbeat rate, and morphological growth defects of the embryo were exhibited.	[73]
*Garcinia hanburyi*/Clusiaceae	Siam gamboge, Hanbury’s Garcinia	Commercial product/dried yellow resin	0.5–1.0 µM	1.76 µM	Gambogic acid triggered a lack of fin developmental in the zebrafish embryo.	[93]
*Garcinia mangostana* (Xanthone crude extract)/Clusiaceae	Mangosteen	Mixture of acetone and water (80:20)	7.81–250 µg/mL	15.63 µg/mL	Among surviving zebrafish, no deformities were found.	[28]
*Geissospermum reticulatum*/Apocynaceae	Challua caspi (Kichwa language)	Ethanol/bark	0.1 µg/mL	n.a.	This plant was nontoxic and caused no deformations to zebrafish, even at high concentrations.	[94]
*Himatanthus drasticus*/Apocynaceae	Janaguba milk (latex)	Commercial product	500–1500 µg/mL	1188.54 µg/mL	This plant can be considered to have low toxicity. No teratogenic effect was observed.	[95]
*Hylocereus polyrhizus*/Cactaceae	Pitaya	Ethanol; water solution (70:30, v/v)/peel & pulp	100–1000 µg/mL and 20 µL)	>1000 µg/mL (96 h)	The results indicated the pulp and peel of this plant were non-toxic.	[96]
*Leonurus japonicus*/Lamiaceae	Oriental/Chinese motherwort	Essential oil/the aerial parts	6.25–100 µg/mL	1.67 ± 0.23 µg/mL	Motherwort essential oil was toxic to embryos. Morphological abnormalities: partial or complete absence of eye development, yolk sac oedema, curved spine, tail deformities, scattered haemorrhages in the oedematous yolk sac, incomplete heart development, and pericardial oedema.	[97]
*Ligusticum chuanxiong*/Apiaceae	Szechuan lovage	A protein-containing polysaccharide/rhizome	0–800,000 µg/L	965 µg/mL	No significant morphological anomalies were found.	[98]
*Maerua subcordata* (Gilg) DeWolf/Capparaceae		Methanol/fruit, leaf, root tuber, and seed	150,000–1,500,000 µg/L	209,000 µg/L	≤5% sub-lethal abnormalities (signs of malformation of the heart).	[4]
*Marsdenia tenacissima* (Xiaoaiping extract)/Asclepiadaceae	Murva (Hindi), Tong-Guan-Teng (Chinese)	Commercial product	0–3200 µg/mL	2660 µg/mL (24 hpf) 2310 µg/mL (48 hpf) 1920 µg/mL (72 hpf) 1910 µg/mL (96 hpf) 1790 µg/mL (120 hpf)	Severe malformation such as impairment of the swim bladder, retaining the yolk, pericardial oedema, and tail curvature. Histopathological study: Xiaoaiping induced lesion of the liver, muscle, and heart.	[99]
*Millettia pachycarpa*/Fabaceae	Bokol-bih, Holsoi, Bokoa-bih (Assamese or Hindi)	Aqueous/root	1–7.5 µg/mL	4.276 µg/mL	There were a few developmental anomalies, such as yolk sac oedema, curvature of the spinal cord, pericardial oedema, swim bladder swelling, reduced heart rate, and late hatching. This plant extract is also linked to reactive oxygen species and apoptosis, which induced embryonic mortality and toxicity (identified in trunk, brain, and tail).	[6]
*Momordica charantia*/Cucurbitaceae	Bitter melon, bitter gourd, bitter apple	Methanol/fruit, seed	1–400 µg/mL	50 µg/mL (only seed extract)	Although it affected the heartbeat, the fruit extract is considered to be harmless and no mortality was observed. The seed extract demonstrated a slight level of developmental failure and had significant cardiac toxicity.	[100]
		Hot and cold aqueous/leaf	15.625–1000 µg/mL	144.54 µg/mL: Hot aqueous Chinese extract 199.53 µg/mL: Hot aqueous Indian extract 251.19 µg/mL: Cold aqueous Chinese extract	For all samples the heart beat differed at higher concentrations. Also, mild toxicity was observed, although consumption of this plant is assumed to be safe.	[12]
		Aqueous/leaf	0.05–3%	3.0% at 24 h	Teratogenic effects such as bent back, tip of tail bent, oedema in the yolk sac, and scoliosis. Delayed development and morphological deformities of the embryo were observed.	[101]
*Moringa oleifera*/Moringaceae	Drumstick tree, Moringa	Hot water/leaf and bark	0.3–6 µg/mL	1.5 µg/mL at 36 h (leaves); 3 µg/mL at 36 h (bark)	Even low concentration was embryo-toxic and had teratogenic impacts on the developing embryos.	[102]
		Essential oil/seed	0.1–1000 µg/mL	21.24 ± 0.44 µg/mL	High concentrations (≥ 50 µg/mL) caused 100% mortality of embryos. The process of angiogenic blood vessel formation in zebrafish embryos was substantially disturbed by the seed oil.	[103]
		Aqueous/seed	12.5–200 µg/mL	190 µg/mL (48 h) 133 µg/mL (72 h) 49 µg/mL (96 h)	No toxic endpoint was seen after 24 h, hatching was prolonged, and the larval period at 96 h compared with the control was decreased.	[104]
*Moringa peregrina*/Moringaceae	Miracle tree	Essential oil/seed	0.1–1000 µg/mL	25.11 ± 0.547 µg/mL	High concentrations (≥ 50 µg/mL) caused 100% mortality of embryos. The process of angiogenic blood vessel formation in zebrafish embryos was substantially disturbed by the seed oil.	[103]
*Olea europaea*/Oleaceae	Olive	Raw and polar fraction (raw oil: commercial)/fruit	1–100%	Raw oil: 11.98% (EC_50_) Polar fraction: 61.87 (EC_50_)	Abnormalities observed included non-hatching, pericardial oedema, and blood accumulation, and more were expressed at a concentration of 11.98%. Raw oil induced 50% of developmental abnormalities compared with the polar fraction. After 96 h, strong developmental retardations such as developmental delay and absence/prolongation of pigmentation formation were observed.	[105]
*Onosma dichroantha*/Boraginaceae	n.a.	Cyclohexane, ethyl acetate, methanol/root	0.02–50 µg/mL	n.a.	n.a.	[106]
*Orthosiphon stamineus*/Lamiaceae	Java tea	Aqueous/whole plant	0–10,000 µg/mL	1685 µg/mL (48 hpf) 1685 µg/mL (96 hpf)	n.a.	[2]
*Palicourea deflexa*/Rubiaceae	n.a.	Methanol (fraction)/leaf	1–100 µg/mL	72.18 µg/mL	At high concentration (100 µg/mL), some possible anomalies, such as no development of somites, decreased pigmentation, and formation of oedema were observed. Lethality only appeared after 96 h.	[52]
*Palmaria palmata*/Archaeplastida	Dulse, dillisk (dilsk), red dulse, sea lettuce flakes (creathnach)	Crude protein (hydrolysate)/leaf	1–10,000 µg/mL	n.a.	At high concentration (10 mg/mL), larvae started developing spinal curvature and a deformed yolk sac. At 5 mg/mL, larvae appeared to have a swollen yolk sac and developed a curved spine.	[51]
*Passiflora caerulea*/Passifloraceae	Bluecrown Passionflower	Aqueous/leaf	40–120 µg/mL	80 µg/mL (40–120 µg/mL)	Mortality increased based on concentration and delay in hatching. No apparent signs of abnormal growth or morphology were detected in embryos at 96 hpf.	[107]
*Peucedanum alsaticum*/Apiaceae	n.a.	n-heptane, ethyl acetate, methanol, water/fruit	50–400 µg/mL	n.a.	n.a.	[33]
*Phyllanthus niruri*/Phyllanthaceae	Gale of the wind, stonebreaker	Aqueous/leaf	0.05–10%	–	Tail malformation and coagulation were detected as the most remarkable toxic effect from the extract; dose-dependent impacts on the heartbeat and hatchability of the embryo also were detected.	[108]
*Piper betle*/Piperaceae	Betel, Ikmo, Daun sirih (Malay), Paan (Hindi)	Hot water/leaf	50–10,000 µg/mL	–	The plant extract affected the hatchability of embryos and was linked to delayed development. Also, a coagulated embryo and tail malformation were observed as a teratogenic effect.	[81]
*Polygonum multiflorum*/Polygonaceae	Tuber fleeceflower	n-hexane, ethyl acetate, water/root	0–175,000 µg/L	TIs: 1430 (ethyl acetate), 630 µg/L	40 and 105 mg/L concentrations (all extracts) induced yolk sac oedema (heart oedema), hemovascular defects, abnormal trunk, and necrosis. The water extract induced synthesis of melanin in zebrafish through stimulation of tyrosinase.	[109]
		Water, ethanol, methanol, acetone/dried roots	0–10,000 µg/mL	129.4 ± 2.7 µg/mL (water) 58.2 ± 1.9 µg/mL (30% ethanol) 39.8 ± 0.8 µg/mL (50% ethanol) 27.9 ± 2.3 µg/mL (70% ethanol) 25.5 ± 2.0 µg/mL (95% ethanol) 48.8 ± 2.8 µg/mL (methanol) 23.6 ± 2.2 µg/mL (acetone)	Teratogenic effects included coagulated embryos, lack of development of the somite, absence of tail detachment and heartbeat, and malformation of the notochord	[110]
*Psoralea corylifolia*/Fabaceae	Babchi	Commercial product (Psoralen)	0–6.70 µg/mL	3.40 µg/mL (LC_50_) 2.52 µg/mL (LC_10_) 1.98 µg/mL (LC_1_)	Psoralen therapy caused hatching rate and body size to decrease and the abnormality rate of zebrafish to increase significantly. Yolk retention, pericardial oedema, swim-bladder malformation, and curved body form were observed.	[111]
*Punica granatum*/Lythraceae	pomegranate	Ethanol/peel	100–250 µg/mL	196,037 ± 9.2 µg/mL	No issues with the reproductive organs, heart, and androgen hormones were detected.	[112]
*Rhizosphora apiculata*/Rhizosphoraceae	Mangrove	Methanol/bark, leaf, stem and root	50–100 µg/mL	n.a.	Higher concentrations of the extract from bark, leaf, and stem did not cause mortality.	[78]
*Salvia miltiorrhiza*/Lamiaceae	Red sage	Commercial product/Tanshinone IIA (diterpene quinone)/root	0.44–7.06 µg/mL	5.44 µg/mL (72 hpf) 3.77 µg/mL (96 hpf)	Tan-IIA showed potential cardiotoxicity and growth inhibition in zebrafish embryos. Scoliosis, tail deformity, and pericardium oedema were the primary signs of teratogenicity.	[45]
*Sida acuta*/Malvaceae	Wireweed	Hexane, chloroform, acetone, methanol/n.a.	10–100 µg/mL	20.9 µg/mL	The methanolic extract led to heartbeat rate reductions (more significant than nebivolol as a positive control). This study reported no teratogenic effects.	[86]
*Solidago canadensis*/Asteraceae	Canada goldenrod, Canadian goldenrod	Ethanol/leaf	0–500µg/mL	0.42 ± 0.03 µg/mL (24 h) 0.33 ± 0.04 µg/mL (48 h) 0.32 ± 0.03 µg/mL (72 h)	This paper only mentioned that low toxicity on zebrafish was observed.	[113]
*Sonneratia alba*/Lythraceae	Mangrove tree	Methanol/bark and leaf	50–100 µg/mL	–	–	[78]
*Spilanthes acmella*/Asteraceae	Paracress, toothache plant, Sichuan buttons, buzz buttons, tingflowers, electric daisy Kradhuawean (Thai)	Aqueous/leaf	0.01–20%	n.a.	No mortality was observed in the highest concentration test.	[89]
*Spondias mombin*/Anacardiaceae	Yellow mombin, hog plum	Hydroethanolic/leaf	1–9 g/kg (LD_50_); 25,000–75,000 µg/L (LC_50_)	4.515 g/kg (LD_50_: 48 h: immersion) 49.86 µg/mL (LC_50_: 48 h: oral)	There were no recorded teratogenic effects.	[114]
*Streblus asper*/Moraceae	Siamese rough bush, khoi, serut, thoothbrush tree	Methanol/bark	50–100 µg/mL	2,000,000 µg/mL	Slight oedema of heart muscles at 100 µg/mL. No other severe malformations were observed.	[115]
*Sutherlandia frutescens*/Fabaceae	Cancer bush, balloon pea, sutherlandia	Aqueous and 80% ethanol	5–50 µg/mL	30.00 µg/mL	Both extracts showed bleeding and pericardial cyst development at high concentration, but the aqueous extract was less toxic to larvae.	[8]
		Aqueous, ethanol/aerial part	5–300 µg/mL	297.57 µg/mL (aqueous) 40.54 µg/mL (ethanol)	The high concentration of both extracts inhibited hatching rate and mortality.	[116]
*Terminalia chebula*/Combretaceae	Myrobalan	Ethanol/fruit, leaf, root	n.a.	n.a.	At 50 µg/L, fish treated with this extract came back to normal condition and showed a related decrease in superoxide dismutase and decreased mitochondria and cell viability.	[74]
*Tetrapterys (Melanolepis) multiglandulosa*/Malpighiaceae	Pakalkal (Tagalog)	Dichloromethane, hexane, ethyl acetate, methanol and water/leaf	n.a.	200 µg/mL (methanol extract)	A significant ecdysteroid was the most toxic in the zebrafish.	[117]
*Tinospora cordifolia*/Menispermaceae	Heart-leaved moonseed, gaduchi, guduchi, giloy, Makabuhay (Filipino)	Aqueous/leaf and bark	0.01–10%	1% (leaves), 10% (barks)	Abnormalities of head and tail, delayed development, reduced mobility, stunted tail, and scoliosis/flexure were observed.	[118]
*Thuja orientalis*/Cupressacae	Oriental Arbor-vitae	Ethanol/leaf	150–2400 µg/mL	702.9 µg/mL	At 2.4 mg/mL embryos were coagulated; at 1.2 mg/mL they did not hatch and oedema was observed; 0.6 mg/mL resulted in skeletal deformities; at 0.3 mg/mL slight oedema was observed; at 0.15 mg/mL normal growth was observed.	[50]
*Trapa natans*/Lythraceae	Water chestnut, water caltrop	Acetone, methanol and ethyl acetate/leaf	100,000–1,000,000 µg/mL	50 µg/mL (methanol) 40 µg/mL (acetone) 30 µg/mL (ethyl acetate)	Methanol and acetone extracts did not show toxicity to zebrafish.	[119]
*Tripterygium wilfordii*/Celasteraceae	Thunder god vine, léi gōng téng (Mandarin)	Purchased commercially each compound	1/10, 1/5, 1/2 LC_50_	Taxol: 0.10, 0.21, 0.53 µg/mL; Gambogic acid 0.03, 0.05, 0.13 µg/mL; Triptolide: 0.04, 0.07, 0.18; Auranofin: 0.84, 1.68, 4.21 µg/mL; Mycophenolic acid: 0.44, 0.95, 2.22; Curcumin: 0.43, 0.87, 2.17 µg/mL; Thalidomide: 0.71, 1.43, 3.56 µg/mL	From those 7 compounds, gambogic acid followed by taxol showed high anticancer activity	[120]
*Zanthoxylum sp.*/Rutaceae	n.a.	Commercial product; zanthoxylum	0–1000 µg/mL	81.18 µg/mL	No teratogenic effects were reported.	[121]

Days post-fertilization: dpf; Hours post-fertilization: hpf; n.a.: not applicable.

**Table 4 plants-09-01345-t004:** Description of toxicological aspects of various studies.

Exposure Format	Number	Exposure Stage	Exposure Time	Parameters Evaluated (Endpoint)	Compound(s)	Reference
96-well plates	1/well	24 hpf	96 h	Embryotoxicity	Armatamide, rubecenamide, lemairamin, rubemamine, and zanthosine	[121]
	1/well (10/group)	6 hpf	96 h	Embryotoxicity and teratogenic effects	Palmitic acid, phytol, hexadecanoic acid, 1-monopalmitin, stigmast-5-ene, pentadecanoic acid, heptadecanoic acid, 1-linolenoylglycerol, and stigmasterol	[84]
	30/group (90 embryos)	n.a.	96 h	Teratogenic effects and embryotoxicity	Alkaloids, flavonoids, phenolics, and saponins	[73]
	1/well	24 hpf	72 h	Acute toxicity assay	Polysaccharides, peptides, protein, lipids, terpenoids, saponins, phenolics, and sterols	[12]
	12/group	2 hpf	96 h	Embryotoxicity and teratogenic effects	α/β-thujone, thujone	[50]
	n.a.	1 hpf (embryo)	72 h	Cytotoxicity and embryonic toxicity test (survival rate)	Xanthone	[28]
	4/group (per well)	12 hpf	48 h	Embryotoxicity (mortality, hatchability, heartbeat rate, and malformation), teratogenic activity	Ascorbic acid, flavonoids, phenolics, and carotenoids	[102]
	1/well	24 hpf	96 h	Embryotoxicity and teratogenic effects	Capsaicin, ethyl palmitate, ethyl linoleate, dihydrocapsaicin, and docosenamide	[84]
	36/group	2.5–3.7 hpf	72 h	Developmental toxicity and behavioural safety	Roots: glycoside, siaraside (pregnane glycoside); Stem bark: α-amyrin, acetate, lupeol, and β-sitosterol	[115]
	12/group	4 cell embryo stage	144 h	Embryotoxicity (heart rate and hatching time)	Spilanthal, and flavonoid	[89]
24-well plates	12 fertilized eggs/group	6 hpf	120 h	Embryotoxicity and teratogenic effects (hatching rate, heartbeat, otolith, blood circulation, tail detachment, and motility)	Catechin, epicatechin, and naringenin	[1]
	20/concentration	6 hpf	96 h	Embryotoxicity		[76]
	10/well	6 hpf	96 h	Embryotoxicity	Methanol extract: phenolic compound (p-heydroxybenzoic acid), phenolic acid, flavonoid, and flavonoid glycoside	[119]
	10/group (30 per tested sample)	64-cell (egg)	96 h	Embryotoxicity	Phenolic compounds: tyrosol, catechol. Hydroxytyrosol, gallic acid, and ascorbic acid.	[105]
	1/well	4 hpf	96 h	Embryotoxicity	Pyrrolizidine alkaloids such as senecionine or senecivernine; phenolic compounds, terpenoids, and iridoids, glucosinolates (glucolepidin and glucobrassicin), alkaloids or amines (stachydrine and trigonelline)	[4]
	4/group	12 hpf	48 h	Embryotoxicity, teratogenic effects	n.a.	[108]
	30/group	4 hpf	96 h	Embryotoxicity	Phenols, and polyphenols	[107]
	10/well	16 cell stage	72 h	Embryotoxicity	n.a.	[82]
	6/well	24 hpf	96 h	Embryotoxicity	Sutherlandin, and sutherlandioside	[116]
	6/group (per well) (n =18)	24 hpf	216 h	Developmental toxicity and cardiotoxicity	No pure compound isolated	[8]
	15/well	2 hpf	96 h	Embryotoxicity, teratogenic effects	Alkaloids (stachydrine)	[97]
	4/well	–	48 h	Embryotoxicity, teratogenic effects	n.a.	[88]
	n.a.	6 hpf	96 h	Acute toxicity in vivo (zebrafish), *in silico* (computational chemistry)	Brevifolin, tannin, saponin, quinone, steroid/triterpenoid, and flavonoid	[112]
	15/well	4 hpf	96 h	Embryotoxicity, teratogenic effects	Psoralen	[111]
	40/group	16-cell stage	96 h	Zebrafish embryo acute toxicity (ZFET)	Emodin, rhein, physcion (dihydroxyanthraquinone), chrysophanol, aloe-emodin, rhaponticin, polygonumnolide, and 2,3,5,4’-tetrahydroxystilbene-2-O-β-D-glucopyranosede, and resveratrol-2-O-β-D-glucopyranoside	[110]
	30/group (10/well)	9 hpf	55 h	Anti-melanogenic and embryotoxicity	Lupeol (triterpenoid), hydrolysable, and 3,3′-di-*O*-methyl ellagic acid (bark of *S. alba*)	[78]
	1/well (20 embryos per concentration)	16-cell stage	168 h	Embryotoxicity	n.a.	[104]
	20 eggs/group (1 embryo per well	1 hpf (embryo)	96 h	Embryotoxicity, teratogenic effects (body axis, head, tail, blood circulation, eyes, heart, pigmentation, somite, and yolk sac)	Curcumin, desmethoxycurcumin, and bisdemethoxycurcumin	[43]
	1/well	32-cell stage	96 h	Embryotoxicity	harman-3-carboxylic acid, βc alkaloids: β-carboline alkaloid	[52]
	30 embryos/well	2 hpf	96 h	Toxicity on cell and zebrafish (coagulation of eggs, tail detachment, presence of heartbeat, and hatching rate)	Alkanes, alkenes, amino acid, amines, and amides	[2]
	20/well	6 hpf	72 h	–	3-O-propionyl-5, 10, 14-O-triacetyl-8-O-(20 -methyl- butanoyl)-cyclomyrsinol, and mirsinance type diterpene	[91]
	20/well	2 hpf	70	Embryotoxicity, teratogenic effects	Gambogic acid	[93]
	5/group (per well)	6 hpf	120 h	Developmental toxicity and apoptosis induction	Rotenone, and saponin	[6]
	10/well	24 hpf	96 h	Embryotoxicity	n.a.	[86]
	6/well	7 dpf	24 h	Determination of maximum tolerated concentration (MTC)	ar-turmerone, α, β-turmerone, and α-atlantone	[85]
12-well plates	4/group (per vial)	1 hpf (embryo)	48 h	Embryotoxicity, teratogenic effects	Beta-momorcharin	[101]
	20/group	6 hpf	96 h	Embryotoxicity, teratogenic effects	Quinochalcones, and flavonoids	[83]
	4/well	–	48 h	Embryotoxicity, teratogenic effects	Flavonoids, terpenoids, cardiac glycosides, saponins, and tannins	[81]
	4/group (per well)	12 hpf	48 h	Embryotoxicity, teratogenic effects	Alkaloids, di-terpenoid lactones, glycosides, steroids, sesquiterpenoid, phenolics, and aliphatic	[118]
	3/well	n.a.	48 h	Embryotoxicity and teratogenicity	Rotenone	[53]
	10/well	48 hpf	48 h	Embryotoxicity	Peptides	[51]
	12/group	egg	120 h	Developmental toxicity (mortality, hatching rate, oedema, and malformations)	Alkaloids, flavonoids, terpene derivatives, glycosides, novel diazepine, and squalene	[77]
Glass beaker	24/group	n.a.	96 h	Acute toxicity test	Phyllocactins, betanins, 2′-*O*-apiosyl-malonyl-betanin, 6′-*O*-malonyl-2-descarboxybetanin, flavonoids, isorhamnetin triglycosides, quercetin-3-*O*-hexoside, and carbohydrates.	[96]
	15/group	0 hpf	48 h	Acute toxicity test, histopathology	Fatty acids, lupeol, and terpenes	[114]
Glass Petri dish	6/group	adult	96 h	Fish acute toxicity	Alkaloids, phenol, and flavonoids with some chemical groups of alkanes, aromatics, hydroxyls, carbonyls, aldehyde, amines, nitriles, amides and ethers	[78]
	30/Petri dish	6 hpf	72 h	Embryotoxicity	*Moringa oleifera* seed oil: 2,2-Bipydidine,3,3-diol, myristoleic acid, palmitoleic acid, stearic acid, oleic acid, arachidic acid, methyl-9-octadecanoate, adrenic acid, erucic acid, behenic acid, decamethylcyclopentasiloxane, α-tocopherol, γ-tocopherol, β-sitosterol, camphenone, hexahydrofarnesylacetone, linoleic acid, gadoleic/eicosenoic acid *Moringa peregrina* seed oil: palmitic acid, stearic acid, oleic acid, arachidic acid, gadoleic/eicosenoic acid, tricosane, erucic acid, behenic acid, α-tocopherol or γ-tocopherol, α-sitosterol, δ-tocopherol, 2-allyl-5-t-butylhydroquinone, isamoxole, hexahydrofarnesylacetone, linoleic acid, and β-sitosterol	[103]
	n.a.	8-cell stage	5 dpf	Embryotoxicity, teratogenic effects	1,2-cyclopentanedione, 2,3-dihydro-3,5-dihydroxy-6-methyl-4h-pyran-4-one, 1,3;2,5-dimethylene-l-rhamnitol, elemol, (-)-selina-4.alpha.,11-diol, and beta-eudesmol	[100]
	20/Petri dish	1–4 cell stage	72 h	Embryotoxicity	Shikonin, β,β-dimethylacrylalkannin, and β,β-dimethylacryl shikonin	[106]
Rectangular glass tank	12/group	n.a.	48 h	Acute toxicity	Fatty acids, esters, alcohols, and the sterol sitosterol	[114]
REKO glass (artificial egg environment)	20 fertilized eggs	4 hpf	48 h	Embryotoxicity, teratogenic effects	Phenolic compounds	[94]
	10/glass tank	n.a.	48 h	Evaluation of genotoxicity and acute toxicity	Phenolic compounds, alkaloids, saponins, terpenoids, carbohydrates, and tannins	[10]
Vial	4/group (per well)	n.a.	48–72 h	Cytotoxicity, toxicity, teratogenicity (hatchability and heartbeat rate)	n.a.	[87]
Vessel	10/group	adult	72 h	Fish acute toxicity	n.a.	[113]

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
