# Peer review of "Zebrafish as a Successful Animal Model for Screening Toxicity of Medicinal Plants"

_plants, 2020, doi:10.3390/plants9101345_

Round 1

Reviewer 1 Report

Modarresi and colleagues present a coherent review on the current state of zebrafish as a model for studying the toxicity of plant extracts on animals. Toxicity studies on animals are outside my area of expertise, so I cannot comment on completeness. Generally, this review is well written, coherent and my impression is that it provides a good overview of toxicity studies on plant extracts using the zebrafish model, while reinforcing the importance of toxicity studies for plant extracts intended for use in humans.

The authors begin by highlighting the advantages of the zebrafish system in comparison to other animal models and position it in relation to other models (tab1) and highlight the potential shortcomings of other model systems. They then present the history of the zebrafish model and introduce different toxicity scoring systems and bioassays. The authors provide ample evidence for the use of Zebrafish in literature in tables 2 and 3. 

The authors then proceed to presenting a number of different parameters that can be assessed using the zebrafish model, e.g. malformation or cardiotoxicity.

Overall the review is clear and follows a logical structure and seems comprehensive. Some language editing is recommended, as some sentences are a bit unclear. Some are given below.

Minor comments:

The meaning of last sentence of the abstract is not clear to me.

l219: "Zebrafish was designed to [...]" reads a bit creationist. I suggest rephrasing to "Zebrafish was found to"

l340: What is the total score?

l395: "The most effective and earlier expression of EGFP showed zebrafish gills[...]" this sentence is not clear to me, it may be that the experiment this sentence references needs to be explained with some more details.

Reviewer 2 Report

Line 4

Who is the corresponding author?

Line 10

At the end there is a quote (... Candidates')

Line 32

(... plants')

Line 33-34

Solvents sort based on polarity or increased use of solvents in Zebrafish tests.

Line 51

Separate (... tests -) please standardize

Review and correct the text the following symbol (') completely, I do not understand its repetitive use.

In Table 1. I suggest placing the consulted references.

In table 3, column of "LC50 (μg/mL)", describe the units and below it describes another unit (mg/mL) to homogenize performing the respective conversion. Do the same with "Concentrations range"

Separate unit values throughout the text (example: 24h) as well as “1.67 ± 0.23”

Change the unit from "hrs" to "h" in the manuscript

In table 3, it is irrelevant to place the names of the countries

In Table 3, is the name of the following compound correct? (-)-selina-4.alpha

Check in all the text of the tables the correct writing of “. . . phenolics, saponins” should be (... phenolics, and saponins)

In table 3, check the correct name of the chemical compound "monotepene thujone", to be more specific "steroid / triterpenoid", in "triterpene lupeol" it should only be "lupeol", you should check in the whole manuscript that when the name of the Only that compound should be placed and not the name of the group to which the compound belongs.

Add footnote to table 3, for example, from: "hours post-fertilization (hpf)"

Correct references according to the author's instructions

References should be described as follows, depending on the type of work:

Journal Articles:
1. Author 1, A.B.; Author 2, C.D. Title of the article. Abbreviated Journal Name YearVolume, page range.

Books and Book Chapters:
2. Author 1, A.; Author 2, B. Book Title, 3rd ed.; Publisher: Publisher Location, Country, Year; pp. 154–196.
3. Author 1, A.; Author 2, B. Title of the chapter. In Book Title, 2nd ed.; Editor 1, A., Editor 2, B., Eds.; Publisher: Publisher Location, Country, Year; Volume 3, pp. 154–196.

Unpublished work, submitted work, personal communication:
4. Author 1, A.B.; Author 2, C. Title of Unpublished Work. status (unpublished; manuscript in preparation).
5. Author 1, A.B.; Author 2, C. Title of Unpublished Work. Abbreviated Journal Name stage of publication (under review; accepted; in press).
6. Author 1, A.B. (University, City, State, Country); Author 2, C. (Institute, City, State, Country). Personal communication, Year.

Conference Proceedings:
7. Author 1, A.B.; Author 2, C.D.; Author 3, E.F. Title of Presentation. In Title of the Collected Work (if available), Proceedings of the Name of the Conference, Location of Conference, Country, Date of Conference; Editor 1, Editor 2, Eds. (if available); Publisher: City, Country, Year (if available); Abstract Number (optional), Pagination (optional).

Thesis:
8. Author 1, A.B. Title of Thesis. Level of Thesis, Degree-Granting University, Location of University, Date of Completion.

Websites:
9. Title of Site. Available online: URL (accessed on Day Month Year).
Unlike published works, websites may change over time or disappear, so we encourage you create an archive of the cited website using a service such as WebCite. Archived websites should be cited using the link provided as follows:
10. Title of Site. URL (archived on Day Month Year).

Round 2

Reviewer 2 Report

In the columns of table 3; concentration and LC50 (µg/mL); (mg/mL); (mg/L)) some values like (1190) have no units, check and describe the units. From the previous recommendation, I do not homogenize the following (for example): 0.06039 mg / mL is equal to 60.39 μg / mL, if you homogenize all the units to a single unit, your table will be better. In other words, a conversion must be carried out.
I mention that you are not altering any information about the author from which you obtained it, it is simply a conversion, and also now you are the authors of the writing you present.

Table 1. Suggested methods for toxicological analysis (based on papers by Rizzo et al. 2013 and d'Amora et al. 89 2018)

Table 1. Suggested methods for toxicological analysis (based on papers by Rizzo et al. 2013 and d'Amora et al. 89 2018)

Regarding your next answer, then it is a separate quote:
Answer 12: Based on the formula the author mentioned in his article it was C16H26. This formula in PubChem is "1-Hexyladamintane". But due to the author's request by email on the date 9/28/2020, he quoted: "Dear Amir Modarresi,
My colleague in the pharmacology department, who did the GC-MS, told me that these two compounds are completely different. The structure of compounds in GC-MS is automatically identified by software after comparing it with the known structure present in the library. "
We decided to continue the analysis of your library on our manuscript.

In the text of line 359, sodium is misspelled and the NA + and K + inhibitors, and these NA + channels are also superscript.

Author Response

Response to Reviewer 2 Comments

Point 1: In the columns of table 3; concentration and LC50 (µg/mL); (mg/mL); (mg/L)) some values like (1190) have no units, check and describe the units. From the previous recommendation, I do not homogenize the following (for example): 0.06039 mg / mL is equal to 60.39 μg / mL, if you homogenize all the units to a single unit, your table will be better. In other words, a conversion must be carried out.
I mention that you are not altering any information about the author from which you obtained it, it is simply a conversion, and also now you are the authors of the writing you present.

Response 1: I tried to homogenize all units based on µg/mL in Table 3.

Point 2: Suggested methods for toxicological analysis (based on papers by Rizzo et al. 2013 and d'Amora et al. 89 2018)

Response 2: I followed advice, and deleted "based on papers by" from title of Table 1.

Point 3: Regarding your next answer, then it is a separate quote:
Answer 12: Based on the formula the author mentioned in his article it was C16H26. This formula in PubChem is "1-Hexyladamintane". But due to the author's request by email on the date 9/28/2020, he quoted: "Dear Amir Modarresi,
My colleague in the pharmacology department, who did the GC-MS, told me that these two compounds are completely different. The structure of compounds in GC-MS is automatically identified by software after comparing it with the known structure present in the library. "
We decided to continue the analysis of your library on our manuscript.

Response 3: According to author of this paper, we followed their library and their compounds in their paper.

Point 4: In the text of line 359, sodium is misspelled and the NA + and K + inhibitors, and these NA + channels are also superscript.

Response 4: I followed reviewer's comment and use superscript for Na+ and K+.